# GeoLLM: Extracting Geospatial Knowledge from Large Language Models

**Rohin Manvi**[†*]    **Samar Khanna**[†]    **Gengchen Mai**[‡]

**Marshall Burke**[†]    **David Lobell**[†]    **Stefano Ermon**[†]

## Abstract

The application of machine learning (ML) in a range of geospatial tasks is increasingly common but often relies on globally available covariates such as satellite imagery that can either be expensive or lack predictive power. Here we explore the question of whether the vast amounts of knowledge found in Internet language corpora, now compressed within large language models (LLMs), can be leveraged for geospatial prediction tasks. We first demonstrate that LLMs embed remarkable spatial information about locations, but naively querying LLMs using geographic coordinates alone is ineffective in predicting key indicators like population density. We then present GeoLLM, a novel method that can effectively extract geospatial knowledge from LLMs with auxiliary map data from OpenStreetMap. We demonstrate the utility of our approach across multiple tasks of central interest to the international community, including the measurement of population density and economic livelihoods. Across these tasks, our method demonstrates a 70% improvement in performance (measured using Pearson's $r^2$) relative to baselines that use nearest neighbors or use information directly from the prompt, and performance equal to or exceeding satellite-based benchmarks in the literature. With GeoLLM, we observe that GPT-3.5 outperforms Llama 2 and RoBERTa by 19% and 51% respectively, suggesting that the performance of our method scales well with the size of the model and its pretraining dataset. Our experiments reveal that LLMs are remarkably sample-efficient, rich in geospatial information, and robust across the globe. Crucially, GeoLLM shows promise in mitigating the limitations of existing geospatial covariates and complementing them well. Code is available on the project website: https://rohinmanvi.github.io/GeoLLM

## 1 Introduction

Geospatial predictions with ML are widely used across various domains, including poverty estimation (Jean et al., 2016; Chi et al., 2021), public health (Nilsen et al., 2021; Areed et al., 2022; Chang et al., 2022), food security (Nakalembe, 2018), biodiversity preservation (Mai et al., 2023c;b), and environmental conservation (Sofaer et al., 2019). The covariates used in these predictions include geographical coordinates, remote sensing data, satellite imagery, human mobility data (Chang et al., 2022), and phone metadata (Blumenstock et al., 2015; Burke et al., 2019). While having access to quality covariates is essential, it can be challenging due to limited spatiotemporal coverage, high costs, and accessibility barriers (Ball et al., 2017). For example, researchers often use publicly available satellite images to estimate socio-economic indicators like asset wealth, population density, and infrastructure access, because they are free and globally available (Yeh et al., 2020; Robinson et al., 2017; Head et al., 2017). However, model predictive power can be limited due to the fact that important features may not be visible from space.

Large language models (LLMs) have proven to be highly effective foundation models that can be fine-tuned or prompted to perform tasks in various domains including healthcare, education, law,

---

[*]Corresponding author, `rohinm@cs.stanford.edu`

[†]Stanford University

[‡]University of Georgia

```
User:  Describe the address "6th Avenue,
Manhattan, New York" in one detailed
paragraph.

GPT-3.5:  6th Avenue, Manhattan, New
York, is a bustling thoroughfare that
cuts through the heart of this iconic
borough.  It stretches from the southern
tip of Manhattan, where it kisses the
edge of the verdant Washington Square
Park, and winds its way northward,
passing through diverse neighborhoods
like Greenwich Village, Chelsea, and
Midtown.  The avenue is a symphony of
activity, with a constant flow of people,
vehicles, and energy.  Skyscrapers
tower overhead, casting long shadows
that provide momentary respite from the
brilliant sunlight that often bathes the
avenue.  Sidewalks bustle with hurried
commuters, street vendors peddle their
wares, and the distant hum of traffic
blends with the occasional blare of a
taxi horn.  The storefronts that line
6th Avenue are a vibrant tapestry of
businesses, ranging from cozy coffee
shops and quirky boutiques to corporate
offices and renowned eateries.  It's a
place where the essence of New York City,
with its unceasing vitality and diverse
culture, is palpable with every step.
```

```
Basic Prompt:  Coordinates:  (40.76208, -73.98042)

Population Density (On a Scale from 0.0 to 9.9):

Finetuned GPT-3.5:  5.2

_______________________________________________

Our Prompt:  Coordinates:  (40.76208, -73.98042)

Address:  "Calyon Building, 6th Avenue, Manhattan
Community Board 5, Manhattan, New York County,
City of New York, New York, United States"

Nearby Places:
"
0.6 km South-West:  Theater District
0.7 km North:  Columbus Circle
0.7 km East:  Midtown East
0.9 km South-West:  Midtown
1.0 km West:  Hell's Kitchen
1.2 km North:  Lincoln Square
1.3 km South-West:  Garment District
1.4 km South-East:  Turtle Bay
1.4 km South:  Jan Karski Corner
1.4 km South:  Midtown South
"

Population Density (On a Scale from 0.0 to 9.9):

Finetuned GPT-3.5:  9.0
```

(a) GPT-3.5 describing an address.          (b) Prompting with coordinates vs our strategy.

Figure 1: Example prompts and corresponding GPT responses. In fig. 1a we show GPT-3.5 demonstrate its geospatial knowledge by asking it to describe an address. However, in fig. 1b (top) prompting GPT-3.5 with just coordinates and finetuning it on population density is insufficient. We demonstrate our prompting strategy in fig. 1b (bottom) with which a finetuned GPT-3.5 is able solve the task correctly (the expected value is 9.0).

finance, and scientific research (Bommasani et al., 2021; Zhao et al., 2023). This is because LLMs have compressed the knowledge contained in their training corpus, which includes billions or trillions of tokens of data from the internet (Del'etang et al., 2023). Here, we seek to understand whether LLMs possess geospatial knowledge and explore methods to extract this knowledge to offer a novel set of geospatial covariates that can enhance various geospatial ML tasks.

As shown in fig. 1a, a substantial amount of geospatial knowledge in LLMs can be revealed simply by querying them to describe an address. However, extracting this knowledge from LLMs is not trivial. While the most natural interface involves using geographic coordinates like latitude and longitude to retrieve specific location information, this often yields poor results as shown in fig. 1b. The inherent challenge lies in the LLMs' capability to understand and interpret these numeric coordinates in relation to real-world locations, especially when the location is remote or lesser-known.

In this paper, we introduce GeoLLM, a novel method that can efficiently extract the vast geospatial knowledge contained in LLMs by fine-tuning them on prompts constructed with auxiliary map data from OpenStreetMap OpenStreetMap contributors (2017). With our prompting strategy shown in fig. 1b, we can pinpoint a location and provide the LLM with enough spatial context information, thereby enabling it to access and leverage its vast geospatial knowledge. We find that including information from nearby locations in the prompt can improve GPT-3.5's (OpenAI, 2023b) performance by 3.3x, measured in Pearson's $r^2$, over simply providing the target location's coordinates.

We find that popular LLM models such as GPT-3.5 and Llama 2 (Touvron et al., 2023) can be fine-tuned to achieve state-of-the-art performance on a variety of large-scale geospatial datasets for tasks including assessing population density, asset wealth, mean income, women's education and more. With GeoLLM, GPT-3.5, Llama 2, and RoBERTa (Liu et al., 2019), show a 70%, 43%, and 13% improvement in Pearson's $r^2$ respectively over baselines that use nearest neighbors and use information directly from the prompt, suggesting that the models' geospatial knowledge scales with their size and the size of their pretraining dataset. Our experiments demonstrate LLMs' remarkable sample efficiency and consistency worldwide. Importantly, GeoLLM could mitigate the limitations of existing geospatial covariates and complement them well.

We present the following main research contributions:

1. Language models possess substantial geospatial knowledge and our method unlocks this knowledge across a range of models and tasks.
2. Constructing the right prompt is key to extracting geospatial knowledge. Our ablations find that prompts constructed from map data allow the models to more efficiently access their knowledge.

## 2 RELATED WORK

**Identifying Knowledge**  Pre-training instills large amounts of knowledge into language models. Since the introduction of pre-trained language models such as BERT (Devlin et al., 2019), many works try to quantify the amount of knowledge they can encode (Roberts et al., 2020; Jiang et al., 2019) and see if they can be used as knowledge bases (Petroni et al., 2019). Other works have looked for more specific types of knowledge such as commonsense (Feldman et al., 2019), temporal (Dhingra et al., 2021; Liu et al., 2021), biomedical (Sung et al., 2021), numerical (Lin et al., 2020), scale (Zhang et al., 2020), and many others. With the recent introduction of instruction-finetuned LLMs, directly querying knowledge with prompts is a potentially simpler method. In this paper, we focus on showing the quantity and quality of geospatial knowledge in pre-trained language models.

**Knowledge Extraction**  Many works aim to improve knowledge extraction from language models by fine-tuning or prompt tuning. Works on prompt tuning explore mining prompts from the web (Jiang et al., 2019), optimizing prompts in the discrete space of words and tokens (Shin et al., 2020), optimizing prompts in the continuous embedding space (Zhong et al., 2021), and using adapters (Newman et al., 2021). However, prompts are often difficult to craft (Qin & Eisner, 2021; Jiang et al., 2019), and are sensitive to small changes. Fine-tuning language models for downstream tasks has shown to be an effective and simple method to tune and elicit specific knowledge for evaluation (Radford & Narasimhan, 2018; Dong et al., 2019). It is also shown that fine-tuning may result in higher performance gains compared to prompt tuning (Fichtel et al., 2021). Additionally, LoRA and quantization are shown to greatly reduce the computational cost of fine-tuning (Hu et al., 2021; Dettmers et al., 2023). For these reasons, we choose to extract geospatial knowledge with fine-tuning.

**NLP for Geospatial Tasks**  Natural language processing (NLP) has been used for many geospatial tasks with various data sources. Hu et al. (2017) extracted semantics relations among cities by leveraging news articles. Sheehan et al. (2019) utilized Wikipedia to predict poverty. Xu et al. (2016) accessed traffic by using OpenStreetMap data. Signorini et al. (2011); Paul & Dredze (2011) utilized tweets to predict track-levels disease activities. Recently, researchers started to explore LLMs' capability on various geospatial tasks. Mai et al. (2023a) demonstrated the usability of large language models on various geospatial applications including fine-grained address recognition, time-series dementia record forecasting, urban function prediction, and so on. GeoGPT (Zhang et al., 2023) has been proposed as a GPT-3.5-based autonomous AI tool that can conduct geospatial data collection, processing, and analysis in an autonomous manner with the instruction of only natural language. However, both works mainly relied on the pre-trained LLMs and did not explore the possibility of fine-tuning LLMs for a geo-specific foundation model. Deng et al. (2023) developed an LLM in geoscience, so-called K2, by fine-tuning on a geoscience text corpus which has shown remarkable performances on various NLP tasks in the geoscience domain. However, the K2 geoscience language foundation model is still limited to the common NLP tasks such as question answering, summarization, text classification, etc. In contrast, in this work, we conduct a deep investigation into the possibility of extracting geospatial knowledge from LLMs and whether we can use it for various real-world geospatial tasks like predicting population density. By fine-tuning various LLMs, we demonstrate the quantity, quality, and scalable nature of geospatial knowledge contained in LLMs.

## 3 METHOD

Abstractly, we want to map geographic coordinates (latitude, longitude) and additional features to a response variable, such as population density or asset wealth. A naive approach would be to spatially interpolate using the coordinates. However, this is far from ideal, especially for small sample sizes. Features or covariates can help, but such data can be difficult to obtain. Given this challenge, we aim to determine how much LLMs know about these coordinates and how we can use that knowledge to

better predict response variables of interest. In this section, we describe our approach of extracting this knowledge with the fine-tuning of language models with prompts generated using map data.

## 3.1 MINIMUM VIABLE GEOSPATIAL PROMPT

A minimum viable prompt needs to specify the location and the prediction task. Our basic prompt consists of geographic coordinates and a description of the task. The use of geographic coordinates is essential to extracting high-resolution data as they are a universal and precise interface for knowledge extraction. The description comprises the task's name and the scale indicating the range of possible labels. The ground truth label, is also added when training an LLM with unsupervised learning. We present an example of this simple structure in the top part of fig. 1b.

While we find that adding the name of the task helps, more specific information, such as the name of the dataset, generally does not help. This suggests that basic context sufficiently assists LLMs in efficiently accessing the knowledge in its weights.

Since the label has to be specified through text for LLMs, we use classification instead of regression. However, we find that a classification task closer to regression (e.g., 0.0 to 9.9 as opposed to 1 to 5) is beneficial, despite the fact that LLMs use softmax. If the distribution in the original dataset already has an approximate uniform distribution, we simply scale the values to the range from 0 to 9.9 and round to one decimal place. If the dataset is not inherently uniform, we distribute the values evenly among 100 bins to maintain a uniform distribution. Each bin is then associated with the range of values from 0.0 to 9.9.

## 3.2 PROMPT WITH MAP DATA

The prompt described above, however, has a major limitation. Through zero-shot prompting, one can easily find that LLMs struggle to identify the locations of coordinates. Not knowing where the coordinates are located would be detrimental for geospatial tasks. This is why we focus on making sure the prompt contains additional context that helps the model understand where the coordinates are located. We also design our prompt to be descriptive and useful for a range of geographical scopes such as ZIP code areas or a single square kilometer.

Our prompt, shown in the lower part of fig. 1b, consists of two additional components constructed using map data:

1. Address: A detailed reverse-geocoded description, encompassing place names from the neighborhood level up to the country itself.

2. Nearby Places: A list of the closest 10 locations in a 100 kilometer radius with their respective names, distances, and directions.

We use OpenStreetMap (OpenStreetMap contributors, 2017) for our map data as it can be queried for free using various APIs, although any map data can be used. In our testing, map data from Google is of higher quality, but was prohibitively expensive. We generate addresses using reverse geocoding from Nominatim (Hoffmann, 2023) and the names and locations of nearby places from Overpass API (Olbricht, 2023). A comprehensive prompt template and additional prompt example is provided in appendix A.2

## 3.3 FINE-TUNING AND INFERENCE WITH LANGUAGE MODELS

**RoBERTa**  RoBERTa$_{\text{BASE}}$ is a 125 million parameter encoder-only transformer model trained on 160 GB worth of text (Liu et al., 2019). We choose this as it and other encoder-only models like BERT (Devlin et al., 2019) are often used for classification and regression tasks (Bommasani et al., 2021). We fine-tune the model with all of its parameters being trainable. It takes in the entire prompt and outputs a continuous value which we round to the nearest bin.

**Llama 2**  Llama 2 7B is a 7 billion parameter decoder-only transformer model trained on 2 trillion tokens (Touvron et al., 2023). Despite being unsuited for regression tasks, it has the potential to perform well as it has been trained on significantly more data. Similar to how the model was pretrained, we fine-tune Llama 2 using unsupervised learning on the prompts and labels concatenated. We find

| Task | Source | Samples | GPT-3.5 | Llama 2 | RoBERTa | GPT-2 | MLP-BERT | XGBoost-FT | XGBoost | k-NN | Nightlight |
|------|--------|---------|---------|---------|---------|-------|----------|------------|---------|------|-----------|
| Population | WorldPop | 10,000 | **0.78** | 0.73 | 0.60 | 0.26 | 0.55 | 0.53 | 0.48 | 0.36 | 0.68 |
| | | 1,000 | **0.73** | 0.63 | 0.41 | 0.18 | 0.42 | 0.34 | 0.32 | 0.11 | 0.62 |
| | | 100 | **0.61** | 0.43 | 0.00 | 0.00 | 0.09 | 0.14 | 0.19 | 0.02 | 0.54 |
| Asset Wealth | DHS | 10,000 | **0.75** | 0.74 | 0.69 | 0.40 | 0.62 | 0.64 | 0.63 | 0.59 | 0.55 |
| | | 1,000 | **0.71** | 0.62 | 0.55 | 0.23 | 0.51 | 0.46 | 0.48 | 0.44 | 0.49 |
| Women Edu | DHS | 10,000 | **0.64** | 0.61 | 0.58 | 0.29 | 0.52 | 0.55 | 0.55 | 0.54 | 0.33 |
| | | 1,000 | **0.60** | 0.52 | 0.47 | 0.07 | 0.44 | 0.36 | 0.40 | 0.40 | 0.28 |
| Sanitation | DHS | 10,000 | **0.65** | 0.63 | 0.62 | 0.32 | 0.59 | 0.61 | 0.57 | 0.57 | 0.41 |
| | | 1,000 | **0.64** | 0.51 | 0.49 | 0.16 | 0.43 | 0.43 | 0.42 | 0.43 | 0.35 |
| Women BMI | DHS | 10,000 | **0.63** | 0.63 | 0.60 | 0.37 | 0.54 | 0.59 | 0.59 | 0.58 | 0.31 |
| | | 1,000 | **0.58** | 0.56 | 0.50 | 0.09 | 0.45 | 0.43 | 0.46 | 0.50 | 0.21 |
| Population | USCB | 1,000 | **0.70** | 0.50 | 0.34 | 0.17 | 0.33 | 0.24 | 0.27 | 0.21 | 0.45 |
| | | 100 | **0.62** | 0.38 | 0.00 | 0.00 | 0.20 | 0.13 | 0.15 | 0.08 | 0.34 |
| Mean Income | USCB | 1,000 | **0.55** | 0.43 | 0.21 | 0.04 | 0.19 | 0.15 | 0.18 | 0.22 | 0.09 |
| | | 100 | **0.46** | 0.29 | 0.01 | 0.00 | 0.08 | 0.03 | 0.04 | 0.10 | 0.07 |
| Hispanic Ratio | USCB | 1,000 | **0.74** | 0.65 | 0.55 | 0.07 | 0.49 | 0.50 | 0.56 | 0.57 | 0.24 |
| | | 100 | **0.64** | 0.32 | 0.11 | 0.00 | 0.36 | 0.19 | 0.23 | 0.40 | 0.14 |
| Home Value | Zillow | 1,000 | **0.87** | 0.74 | 0.50 | 0.08 | 0.46 | 0.46 | 0.50 | 0.59 | 0.14 |
| | | 100 | **0.74** | 0.43 | 0.00 | 0.00 | 0.35 | 0.13 | 0.27 | 0.40 | 0.06 |

Table 1: Pearson's $r^2$ for all models across all tasks and training sample sizes.

that this unsupervised learning works better than using supervised finetuning. This means that the model is learning to generate the prompt as well as the label. One could imagine that it is beneficial to learn to generate the address given the coordinates or the nearby places using the coordinates and address. During inference time, we provide it with the prompt and let it generate the three tokens required for the prediction (e.g., "9", ".", and "9"). QLoRA (Dettmers et al., 2023) allows us to greatly reduce the computational cost and effectively fine-tune with just 33 million trainable parameters. However, since it also quantizes the frozen model to 4-bit, this comes with the potential cost of diluting the knowledge contained in the weights.

**GPT-2** GPT-2 is a 124 million parameter decoder-only transformer model trained on 40 GB worth of text (Radford et al., 2019). We use the same finetuning and inference process as Llama 2. We use this model as an LLM baseline and as a way to gain insight into the effect of the size of the model and its dataset on performance.

**GPT-3.5** GPT-3.5 (`gpt-3.5-turbo-0613`) (OpenAI, 2023b) is a chat based LLM. It is the successor of GPT-3 which is a decoder-only transformer model with 175 billion parameters (Brown et al., 2020). GPT-3.5 shows promise in containing a substantial amount of knowledge (Bian et al., 2023). We fine-tune GPT-3.5 through OpenAI's fine-tuning API (OpenAI, 2023a). Similar to Llama 2, we provide it the prompt and let it generate the three tokens required for the completion.

The details of the finetuning of these models and their hyperparameters are specified in appendix A.3. We also demonstrate the importance of fine-tuning in appendix A.4 by testing the few-shot performance of GPT-3.5 and GPT-4.

## 4 EXPERIMENTS

We construct a comprehensive benchmark that encompasses a wide range of geospatial prediction tasks. This benchmark includes various domains and geographies, ranging from women's education levels in developing countries to housing prices in the US. We develop a robust set of baselines to demonstrate the performance attainable using the information directly in the prompt. Finally, we present our results and an ablation study.

### 4.1 TASKS AND SOURCES

**WorldPop** WorldPop (Tatem, 2017) is a research group focused on providing spatial demographic data for the use of health and development applications, including addressing the Sustainable Development Goals. We use their population counts dataset that covers the whole world for the population density task. More specifically, we use their unconstrained global mosaic for 2020 that has a res-

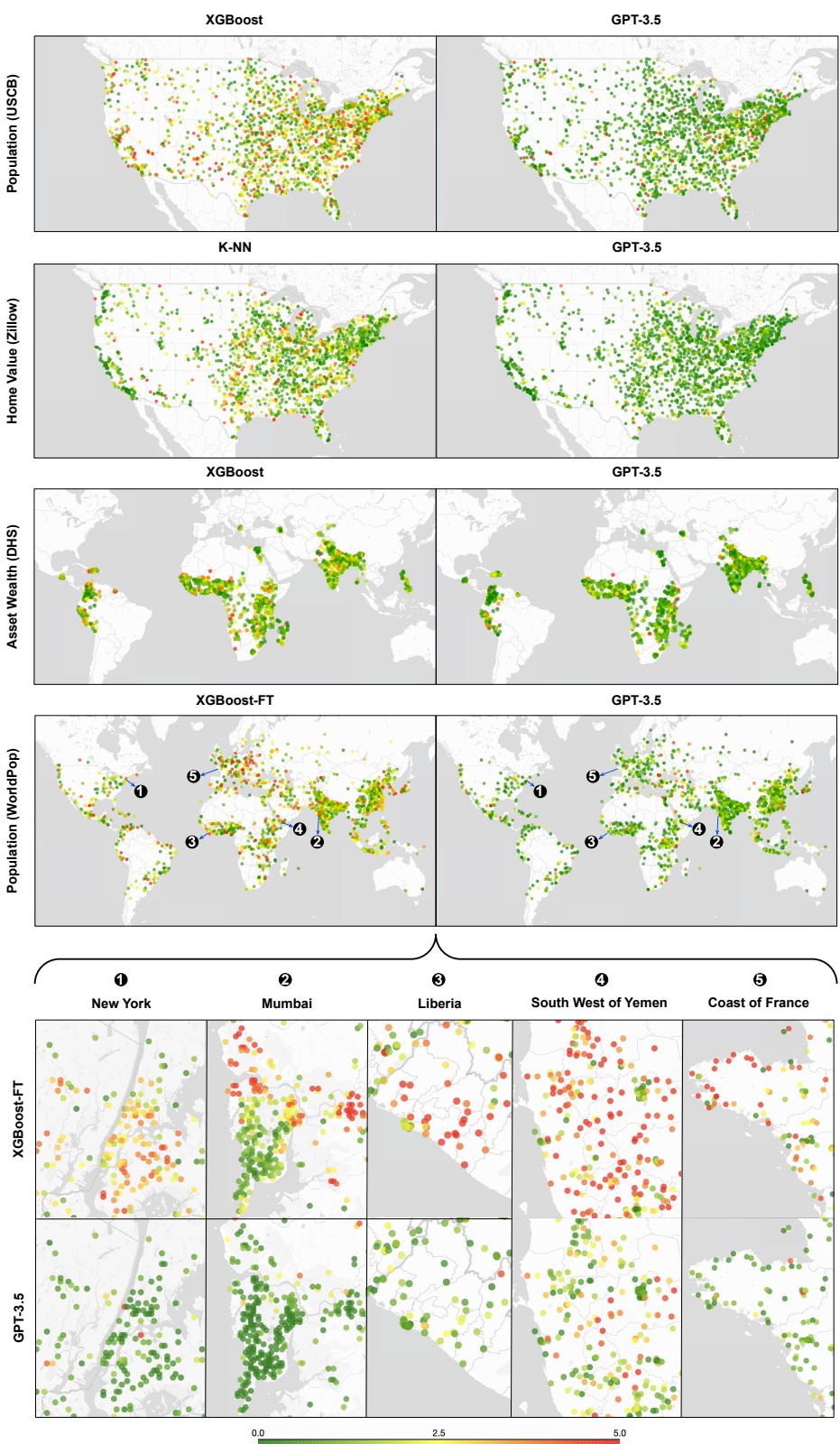

Figure 2: Plots of absolute error comparing the best baselines utilizing no pretraining and GPT-3.5 on tasks from each source with 1,000 samples. We also provide high-resolution plots from various locations around the world for the population density task from WorldPop. We show that GeoLLM not only outperforms baselines on a variety of tasks (table 1) but also demonstrates remarkable geographic consistency across the world.

olution of 30 arc (approximately 1km at the equator) (WorldPop & CIESIN, 2018). To ensure a comprehensive representation of populations, we employed importance sampling, weighted by population. This allows us to capture a wide range of populations; without it, our samples would largely consist of sparsely populated areas. To construct a learning curve for this task, we use training sample sizes of 100, 1,000, and 10,000.

**DHS** SustainBench (Yeh et al., 2021) includes tasks derived from survey data from the Demographic and Health Surveys (DHS) program (DHS, 2023). These surveys constitute nationally representative household-level data on assets, housing conditions, and education levels, among other attributes across 48 countries. While DHS surveys provide household-level data, SustainBench summarizes the survey data into "cluster-level" labels, where a "cluster" roughly corresponds to a village or local community. The geocoordinates are "jittered" randomly up to 2km for urban clusters and 10km for rural clusters to protect survey participant privacy. From their datasets, we use asset wealth index, women educational attainment, sanitation index, and women BMI. Since their datasets cover a wide scope, we use training sample sizes of 1,000 and 10,000.

**USCB** The United States Census Bureau (USCB) (US, 2023) is responsible for producing data about the American people and economy. In addition to the decennial census, the Census Bureau continually conducts annual demographics surveys through the American Community Survey (ACS) program. From these two sources, we get tasks for population, mean income, and Hispanic to Non-Hispanic ratio by what they call ZIP Code Tabulation Areas. Since their data only covers the US, we use training sample sizes of 100 and 1,000.

**Zillow** Zillow provides the Zillow Home Value Index (ZHVI) (ZR, 2023): A measure of the typical home value and market changes across a given region and housing type. It reflects the typical value for homes in the 35th to 65th percentile range. We use the smoothed, seasonally adjusted measure that they provide. Like with the USCB data, we also use this data on a ZIP code level. This serves as our home value task. Since this dataset also only covers the US, we use training sample sizes of 100 and 1,000.

When explicit coordinates are not given in the datasets, we approximate them by determining the center of relevant areas. For instance, we approximate the center of each of the ZIP code area so that we can generate prompts for the tasks from USCB or Zillow. We evaluate performance using the squared Pearson correlation coefficient $r^2$ between the predictions and labels so that comparisons can be made with previous literature (Perez et al., 2017; Jean et al., 2016; Yeh et al., 2020; 2021). We use 2,000 samples for our test and validation sets across all tasks. We split the data into training, test, and validation partitions early in the process before sampling different sizes. We also explicitly check again for any overlap between the train and test samples before fine-tuning or training.

## 4.2 BASELINES

The purpose of the first three baselines is to demonstrate the performance attainable using the information in the prompt. Pre-trained language models can surpass these baselines only if they effectively use the knowledge embedded in their weights. This helps us understand the extent of knowledge encapsulated within pre-trained language models. The next baseline has a pretrained component but provides further insight into performance attainable using sentence embeddings. The last baseline allows for comparison with a well-established satellite-based method.

**k-NN** Our simplest baseline is k-NN. It takes the mean of the 5 closest neighbors. This is a reasonable baseline as physical distance is very relevant for geospatial tasks.

**XGBoost** Our second baseline uses XGBoost (Chen & Guestrin, 2016). This baseline incorporates all numerical and categorical data from the prompt, including coordinates, distances, and directions. It could outperform k-NN since it leverages not just the coordinates, but also the distance and direction data. We established this baseline believing that the direction and distance information could be covariates for some of the tasks we use.

**XGBoost-FT** Our third baseline, XGBoost-FT, leverages both XGBoost and fastText (Joulin et al., 2016). This baseline uses all information contained in the prompt. In addition to using the coordinates, distances, and directions, it also uses embeddings generated from the address and the names of the nearby places. We use fastText to learn sentence embeddings on any given dataset using strings created by concatenating the address and the names of the nearby places.

**MLP-BERT**   Our fourth baseline, MLP-BERT, also uses coordinates, distances, directions, and an embedding of the address and the names of the nearby places. Unlike XGBoost-FT, it uses sentence embeddings from the 110 million parameter pre-trained BERT (Devlin et al., 2019). To accommodate for the larger embedding size, we use a 2-layer MLP.

**Nightlight**   Our final baseline uses publicly available satellite images of luminosity at night ("nightlights") from VIIRS which comes at a resolution of 500 meters (Elvidge et al., 2017). Nightlight images are commonly used as they are a proxy for economic development (Yeh et al., 2020; Sheehan et al., 2019; Jean et al., 2016). We find that images of sizes 16 km by 16 km work best for our diverse set of tasks. We use GBTs as they have been shown to be effective for single-band nightlight imagery (Perez et al., 2017). We find that ResNet (He et al., 2015) models, as used in (Yeh et al., 2020), require large numbers of samples to outperform GBTs.

All baselines except GPT-2, treat the tasks as a regression problem. Their output is rounded to the nearest bin (0.0 to 9.9) for classification. We get the average of the values in a 25 square kilometer area. We then use those values to get the ground truth ranking that is used in the experiments.

### 4.3   PERFORMANCE ON TASKS

From the results presented in table 1 and table 4, it is evident that LLMs are the best-performing models. Among them, GPT-3.5 stands out as the best-performing LLM. Its Pearson's $r^2$ ranges from 0.46 to 0.74, 0.55 to 0.87, and 0.63 to 0.78 for sample sizes of 100, 1,000, and 10,000, respectively. While these variations indicate that the tasks differ in difficulty, LLMs generally outperform all prompt-based baselines and even the nightlight baseline. This suggests that LLMs possess a substantial amount of geospatial knowledge.

Not only does GPT-3.5 outperform all other models on every test, but its performance is also relatively consistent across tasks and sample sizes. Llama 2 does better than all baselines for 17 out of the 19 total tests and consistently does better than RoBERTa. RoBERTa consistently does better than all prompt-based baselines at 10,000 training samples, but struggles at lower sample sizes. Furthermore, GPT-3.5 and Llama 2 retain a much more acceptable level of performance at a sample size of 100, emphasizing their sample efficiency.

GPT-3.5 and Llama 2 do especially well with the population tasks from WorldPop and the USCB compared to the baselines. GPT-3.5 is also especially impressive with the home value task from Zillow with a Pearon's $r^2$ of up to 0.87. However, the difference in performance between the models is less pronounced for the tasks from the DHS. This might be due to the noise that is added when the coordinates for these tasks are "jittered" up to 5 kilometers. With the added noise, it is potentially more difficult to achieve a high Pearson's $r^2$.

As shown in fig. 3 GPT-3.5 outperforms Llama 2 and RoBERTa by 19% and 51% respectively, indicating that the method scales well. Fig. 4 again shows that the sample efficiencies of Llama 2 and GPT-3.5 are exceptional, especially when making predictions on a global scale. Note that with larger sample sizes the gaps in performance will decrease as the physical distances between the training coordinates and test coordinates become smaller.

It is visually clear in fig. 2 that GPT-3.5 does substantially better than the best prompt-based baselines trained from scratch for each task. It also appears that GPT-3.5 generally only struggles where the baselines struggle, suggesting that those areas might just be difficult to estimate rather than it being a specific failing of the model. Notably, GPT-3.5 shows has impressive geographic consistency.

### 4.4   ABLATIONS ON THE PROMPT

The ablation study shown in table 2 demonstrates the importance of using a highly descriptive prompt that helps the model understand where the coordinates are located and efficiently access the knowledge contained in its weights. It also suggests that all components of the prompt contribute to the overall performance of the models. In addition to performing better than the other language models, GPT-3.5 is more resilient to the removal of the map data. This is evident from the fact that prompt without map data is completely ineffective for Llama 2 and RoBERTa. This suggests that GPT-3.5 can more effectively access its knowledge.

| Ablation | GPT-3.5 | Llama 2 | RoBERTa |
|---|---|---|---|
| Whole prompt | 0.73 | 0.63 | 0.41 |
| Removed coordinates | 0.72 | 0.63 | 0.36 |
| Removed address | 0.68 | 0.51 | 0.38 |
| Removed nearby places | 0.58 | 0.47 | 0.28 |
| Only nearby places | 0.65 | 0.51 | 0.38 |
| Only address | 0.57 | 0.48 | 0.28 |
| Only coordinates | 0.22 | 0.00 | 0.00 |

Table 2: Pearson's $r^2$ for ablations with the population density task from WorldPop and 1,000 training samples.

## 5 DISCUSSION

LLMs not only match but also exceed the performance of methods that use satellite imagery. Satellite-based methods achieve Pearson's $r^2$ of 0.56 (Jean et al., 2016), 0.71 (Perez et al., 2017), and 0.67 (Yeh et al., 2020) in predicting asset wealth in Africa. With 10,000 training samples (approximately 3,700 of which are from Africa), a fraction of the over 87,000 data points available from the DHS, GPT-3.5 achieves an $r^2$ of 0.72 when evaluated in Africa.

The performance of the LLMs is particularly notable considering an inherent disadvantage they have relative to the baselines and RoBERTa. Generating multiple tokens with softmax is far less intuitive than regression which the other models use. Despite this, there are substantial improvements in performance and sample efficiency with the LLMs.

It also appears that the significant increase in performance is not simply due to the number of parameters being trained during fine-tuning. With QLoRA, Llama 2 has 33 million trainable parameters as opposed to RoBERTa's 125 million trainable parameters. This suggests that the performance increase is due to the knowledge contained in Llama 2's frozen weights.

This method's performance appears to scale with both the size of the model's pretraining dataset and its capacity for data compression, a capability influenced by the model's size. This suggests our method likely scales with the general capability of the model as well (Kaplan et al., 2020). For example, GPT-4 (OpenAI, 2023c) would likely outperform GPT-3.5.

One could potentially use our method to better understand the biases of LLMs that they inherit from their training corpora. Surprisingly, the performance across countries and continents appears very consistent, as evident from the maps of absolute errors shown in fig. 2. However, preliminary evidence of biases towards sparsely populated or underdeveloped areas is shown in appendix A.6.

This work lays the foundation for the future use of LLMs for geospatial tasks. Additional dimensions could extend the applicability of our method. For instance, one could augment prompts with names of amenities or roads to enrich the geospatial context and allow for even higher-resolution knowledge extraction. Temporal data such as dates could be integrated into the prompts for making forecasts. Multimodal LLMs could potentially also leverage satellite imagery or street view images. Overall, there are likely many ways to extend GeoLLM.

## 6 CONCLUSION

In this work, we introduced GeoLLM, a novel method that can efficiently extract the vast geospatial knowledge contained in LLMs by fine-tuning them on prompts constructed with auxiliary map data from OpenStreetMap. We found that including information from nearby locations in the prompt greatly improves LLMs' performance. We demonstrated the utility of our approach across various tasks from multiple large-scale real-world datasets. Our method demonstrated a 70% improvement in Pearson's $r^2$ over traditional baselines including k-NN and XGBoost, exceeding the performance of satellite-based methods. With GeoLLM, we observed that GPT-3.5 outperforms Llama 2 by 19% and RoBERTa by 51%, suggesting that the performance of our method scales well with the size of the model and its pretraining dataset. Our simple method revealed that LLMs are sample-efficient, rich in geospatial information, and robust across the globe. Crucially, GeoLLM shows promise in substantially mitigating the limitations of traditional geospatial covariates.

## 7 ACKNOWLEDGEMENTS

This research is based upon work supported in part by the Office of the Director of National Intelligence (ODNI), Intelligence Advanced Research Projects Activity (IARPA), via 2021-2011000004, NSF(#1651565), ARO (W911NF-21-1-0125), ONR (N00014-23-1-2159), CZ Biohub, HAI. The views and conclusions contained herein are those of the authors and should not be interpreted as necessarily representing the official policies, either expressed or implied, of ODNI, IARPA, or the U.S. Government. The U.S. Government is authorized to reproduce and distribute reprints for governmental purposes not-withstanding any copyright annotation therein.

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

# A APPENDIX

## A.1 ADDITIONAL VISUALIZATIONS OF PERFORMANCE

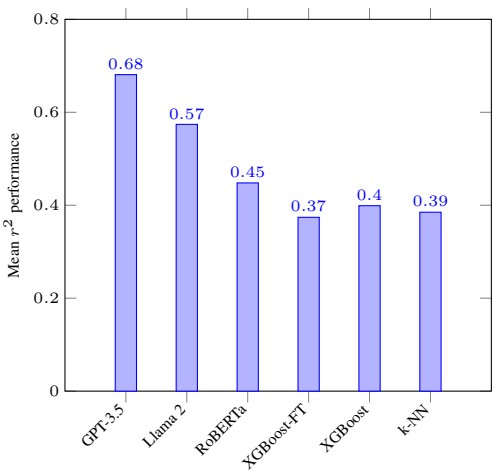

Figure 3: Mean Pearson's $r^2$ for models across all tasks at 1,000 training samples

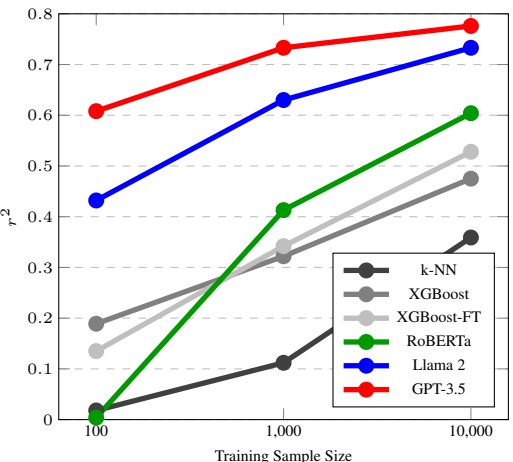

Figure 4: Learning curves for population density task from WorldPop.

## A.2 PROMPT TEMPLATE

To aid the reproducibility of our results, we provide the prompt template, shown in fig. 5a, that is used to construct all prompts across all tasks. In addition to the example shown in the lower part of fig. 1b we provide an example used for the asset wealth task shown in fig. 5b that further clarifies how the prompt template is used.

```
Coordinates:  (<latitude>, <longitude>)

Address:  "<address>"

Nearby Places:
"
<distance> km <direction>:  <place>
<distance> km <direction>:  <place>
<distance> km <direction>:  <place>
<distance> km <direction>:  <place>
<distance> km <direction>:  <place>
<distance> km <direction>:  <place>
<distance> km <direction>:  <place>
<distance> km <direction>:  <place>
<distance> km <direction>:  <place>
<distance> km <direction>:  <place>
"

<task> (On a Scale from 0.0 to 9.9):  <label>
```

(a) Template of a prompt and its label.

```
Coordinates:  (9.63708, 8.48625)

Address:  "Manchok, Kaura, Kaduna, Nigeria"

Nearby Places:
"
4.7 km North-East:  Manchok
9.1 km North:  Jankasa
10.0 km West:  Matagami
12.0 km West:  Kagoro
15.2 km South:  Tukun Kasa
15.7 km South-East:  Manyi
16.3 km South-West:  Tafan
17.1 km South:  Wuro Tabo
18.7 km North:  Rahama
18.9 km South:  Kanafi
"

Asset Wealth (On a Scale from 0.0 to 9.9):  2.1
```

(b) Example of a prompt and its label.

Figure 5: The <latitude> and <longitude> are always rounded to 5 decimal places. The <address> is a reverse geocoded address. The <distance> is always in kilometers and is rounded to one decimal place. The <direction> is one of eight cardinal and intercardinal directions which are "North", "East", "South", "West", "North-East", "South-East", "South-West", or "North-West". The <place> is the actual name of the place. The <task> is the name of the task. In our experiments, the <task> is one of "Population Density", "Asset Wealth", "Women's Education", "Sanitation", "Women's BMI", "Mean Income", "Hispanic/Latino to Non-Hispanic/Latino Ratio", or "Home Value". The <label> is the ground truth (or completion, which is a prediction from an LLM during inference) which can be any number between 0.0 and 9.9 rounded to one decimal place.

## A.3 LANGUAGE MODEL FINETUNING DETAILS

**RoBERTa** We fine-tune the model for regression with all of its parameters being trainable. With a single V100 GPU, finetuning never takes more than 2 hours. We use a learning rate of $1 \times 10^{-5}$ with AdamW optimizer, training for 8 epochs with a batch size of 16, warmup ratio of 0.1, weight decay of 0.1, and a cosine learning rate scheduler.

**Llama 2** Similar to how the model was pretrained, we fine-tune Llama 2 using unsupervised learning with the prompts and labels concatenated. With QLoRA, finetuning never takes more than 2 hours for 10,000 samples with a single A100 GPU. For LoRA, we use rank of 64, alpha of 16, and a dropout probability of 0.1. We also use 4-bit quantization with 4-bit NormalFloat. We train for 4 epochs with bfloat16 mixed precision training, a batch size of 8, gradient accumulation steps of 2, gradient checkpoints enabled, a maximum gradient norm of 0.3, an initial learning rate of $1.5 \times 10^{-3}$ for the AdamW optimizer, a weight decay of 0.001, a cosine learning rate scheduler, and a warmup ratio of 0.03. Interestingly, the model's performance still improves even when the loss on the validation set increases by the fourth epoch. This suggests that the model is less prone to overfitting on the prediction tasks compared to the prompts.

**GPT-2** We use the same finetuning process for GPT-2 as Llama 2. The only difference is that we train the model for 8 epochs instead of 4. Other changes in hyperparameters do not seem to increase performance significantly.

**GPT-3.5** We find that 4, 3, and 2 epochs work well for 100, 1,000, and 10,000 training samples respectively. This costs roughly $0.67, $5, and $33 for those sample sizes. It is worth noting that the specific fine-tuning procedures and internal architecture of GPT-3.5 are not publicly disclosed by OpenAI.

## A.4 FEW-SHOT

We demonstrate GPT-3.5 and GPT-4 few-shot performance by providing them with a series of the 10 closest (by physical distance) training samples for each test example. We use system messages "You are a detailed and knowledgeable geographer" and "You complete sequences of data with predictions/estimates" before providing the 10 examples.

The test set consists of 200 samples. The size reduction of the test set is required as 10-shot is quite expensive, especially with GPT-4. We also reevaluate the performance of the finetuned GPT-3.5 model with this smaller test set to be able to make direct comparisons.

The results in table 3 demonstrate that while few-shot does work, it performs significantly worse than the respective finetuned model as can be seen with the performance difference between the GPT-3.5 models. Interestingly, GPT-4 does far better than GPT-3.5 in the few-shot setting, suggesting that a fine-tuned version of this model could potentially perform significantly better.

| Task | Source | Total Samples | 10-Shot GPT-4 | 10-Shot GPT-3.5 | Finetuned GPT-3.5 |
|------|--------|---------------|---------------|-----------------|-------------------|
| Population | WorldPop | 10,000 | 0.63 | 0.53 | **0.80** |
| Asset Wealth | DHS | 10,000 | 0.65 | 0.62 | **0.78** |
| Population | USCB | 1,000 | 0.49 | 0.39 | **0.73** |
| Home Value | Zillow | 1,000 | 0.79 | 0.55 | **0.86** |

Table 3: Pearson's $r^2$ for few-shot performance.

## A.5 MEAN ABSOLUTE ERROR

As a potentially more interpretable metric, we present the mean absolute error (MAE) for all models across all tasks and training sample sizes in table 4. One can observe that the same conclusions that are made with the Pearson's $r^2$ can be made when comparing the MAE of the various models. In particular, the relative ranking in performance of the models within tasks is consistent with Pearson's $r^2$.

| Task | Source | Samples | GPT-3.5 | Llama 2 | RoBERTa | GPT-2 | MLP-BERT | XGBoost-FT | XGBoost | k-NN | Nightlight |
|------|--------|---------|---------|---------|---------|-------|----------|------------|---------|------|-----------|
| Population | WorldPop | 10,000 | **1.02** | 1.12 | 1.44 | 2.32 | 1.49 | 1.56 | 1.65 | 1.81 | 1.28 |
|  |  | 1,000 | **1.13** | 1.40 | 1.75 | 2.69 | 1.81 | 1.88 | 1.92 | 2.26 | 1.38 |
|  |  | 100 | **1.38** | 2.22 | 2.47 | 3.86 | 2.48 | 2.26 | 2.18 | 2.48 | 1.55 |
| Asset Wealth | DHS | 10,000 | **0.94** | 0.97 | 1.06 | 1.64 | 1.22 | 1.19 | 1.20 | 1.21 | 1.34 |
|  |  | 1,000 | **1.02** | 1.24 | 1.33 | 1.94 | 1.44 | 1.48 | 1.45 | 1.51 | 1.42 |
| Women Edu | DHS | 10,000 | **0.89** | 0.92 | 0.98 | 1.36 | 1.09 | 1.00 | 1.01 | 1.00 | 1.27 |
|  |  | 1,000 | **0.95** | 1.08 | 1.13 | 3.62 | 1.14 | 1.23 | 1.19 | 1.17 | 1.30 |
| Sanitation | DHS | 10,000 | **1.44** | 1.49 | 1.46 | 2.44 | 1.60 | 1.57 | 1.64 | 1.57 | 1.96 |
|  |  | 1,000 | **1.46** | 1.86 | 1.81 | 2.98 | 1.93 | 1.96 | 1.93 | 1.90 | 2.08 |
| Women BMI | DHS | 10,000 | **0.37** | 0.37 | 0.39 | 0.50 | 0.41 | 0.39 | 0.39 | 0.39 | 0.51 |
|  |  | 1,000 | **0.40** | 0.41 | 0.45 | 0.60 | 0.47 | 0.46 | 0.45 | 0.43 | 0.54 |
| Population | USCB | 1,000 | **1.03** | 1.64 | 1.90 | 2.59 | 1.97 | 2.06 | 2.01 | 2.11 | 1.63 |
|  |  | 100 | **1.35** | 1.98 | 2.50 | 3.07 | 2.15 | 2.27 | 2.24 | 2.34 | 1.82 |
| Mean Income | USCB | 1,000 | **1.41** | 1.74 | 2.09 | 3.24 | 2.17 | 2.19 | 2.14 | 2.05 | 2.32 |
|  |  | 100 | **1.61** | 2.41 | 2.44 | 4.28 | 2.36 | 2.48 | 2.53 | 2.38 | 2.48 |
| Hispanic Ratio | USCB | 1,000 | **1.11** | 1.46 | 1.52 | 3.05 | 1.66 | 1.60 | 1.48 | 1.45 | 2.02 |
|  |  | 100 | **1.30** | 2.43 | 2.45 | 3.60 | 1.94 | 2.13 | 2.11 | 1.78 | 2.27 |
| Home Value | Zillow | 1,000 | **0.74** | 1.12 | 1.67 | 2.82 | 1.72 | 1.74 | 1.63 | 1.44 | 2.23 |
|  |  | 100 | **1.11** | 1.81 | 2.51 | 3.26 | 1.94 | 2.29 | 2.03 | 1.81 | 2.44 |

Table 4: Mean absolute error (MAE) for all models across all tasks and training sample sizes.

## A.6 PRELIMINARY INVESTIGATION ON PERFORMANCE BIAS

While there does not appear to be any obvious signs of performance bias across countries or continents as seen in fig. 2, the existence of biases in performance is inevitable as it is very likely that the internet training corpora of LLMs are inherently biased towards developed and densely populated areas. We find preliminary signs of this as the increase in MAE shown in table 5 indicates moderate performance bias towards densely populated and developed areas. However, further research is needed for a comprehensive analysis of the geospatial biases in LLMs and their training corpora. As we have demonstrated here, GeoLLM has the potential to be used as a tool to reveal LLM biases on a geographical scale.

| Task | Source | Above Median Ground Truth | Below Median Ground Truth |
|------|--------|---------------------------|---------------------------|
| Population | WorldPop | 0.94 | 1.09 |
| Asset Wealth | DHS | 0.88 | 0.98 |

Table 5: Mean absolute errors of GPT-3.5 trained on 10,000 samples for population density and asset wealth tasks. Performance is shown for the subsets above and below the median ground truth values (ground truth population density and asset wealth respectively).

