# OpenReview forum: "GeoLLM: Extracting Geospatial Knowledge from Large Language Models"
_ICLR.cc/2024/Conference — ICLR 2024 poster_

### Official Review · Reviewer_92yi · 2023-10-21

**Soundness:** 3 good
**Presentation:** 3 good
**Contribution:** 2 fair
**Rating:** 6
**Confidence:** 4

**Summary:**

This paper explores possible ways to extract and tease out the geospatial information embedded inside the knowledge bases like LLMs. Specifically, they fine-tune LLMs such as chatGPT and Llama-2 with training data curated using OpenMaps with labels derived from various sources for geospatial tasks. On a variety of tasks, they show that fine-tuning LLMs on such data can effectively outperform several baselines. They further establish the geographical consistency of their results as well as ablation on suitable prompts.

**Strengths:**

- This paper reveals an innovative way to use LLMs towards a expert tasks such as computing geospatial metrics without access to various kinds of data.

- The access to various covariates, as mentioned, might not always be available - so the motivation of this work is sound and valid.

- The experiments cover a wide variety of tasks and settings, indicating the power of the proposed approach.

- The experiments involve comparison between multiple LMs like RoBERTa, GPT3.5 and LLama-2, with several useful observations pertaining to performance as well as the data efficiency of each of those.

**Weaknesses:**

- There are no experiments which really delineate where the performance improvements of the final model are coming from. The chosen baselines are not that strong. To delineate the knowledge offered by the LLMs compared to that of the training data, can the authors also conduct an experiment where they fine-tune a normal neural network (not LLM) on the created training data? For example, we can first pass the prompt to a sentence encoder, and then train an MLP/Neural Network which uses this embedding to predict the output as a regression or classification task  (using the same training data used during fine-tuning). This would separate the contributions offered by the LLM compared to the training data which was used in fine-tuning.

-  Adding to the above, can the authors also provide results with using zeroshot capabilities of the LLMs? This would help to separate the importance of fine-tuning on custom prompts.

- Assuming the covariates are indeed available, does the current method offer complimentary benefits to further improve the performance?

- The authors, if possible, should also make comparisons with fine-tuning several other geo-spatial LLM type models, like K2 (Deng et. al. 2023) through fine-tuning on the same set of data.

**Questions:**

Overall, I have several requests to further delineate the performance improvements due to various aspects, while also using stronger baselines as detailed above. I am ready to raise my rating upon satisfactory rebuttal.

---

> ### Author Response · Authors · 2023-11-20
> **Response to Reviewer 92yi**
>
> Thank you for acknowledging the novelty of GeoLLM, the clear use cases and motivation, and constructive criticism. Your suggestions have provided valuable directions for further strengthening our work. Please find our responses to each of your points below:
>
> **Q1: Use of Stronger Baselines**
>
> **A1:** Please see “Stronger baselines” in the overall response. We think that “Full set of ablations” in the overall response may also be helpful.
>
> **Q2: Zero-shot Capabilities of LLMs**
>
> **A2:** Please see “Zero-shot and few-shot experiments” in the overall response.
>
> **Q3: Complementarity with Available Covariates**
>
> **A3:** We added an additional baseline “Nighlight” in Table 1 and Table 3. Please refer to “Stronger baselines” in our general response for the description and discussion of this baseline. The Nightlight baseline provides insights into the baseline performance attainable using a well-established satellite image based method. Notably, LLMs seem to be far more robust on a wide variety of tasks when compared to this baseline. Somewhat predictably, Nightlight is only particularly strong on the population tasks. This supports the idea that LLMs are a richer source of knowledge and a better covariate compared to the well-established nightlight satellite imagery. In practice, GeoLLM, satellite imagery, and other covariates would be used together for the best predictions.
>
> **Q4: Finetuning Other Geo-spatial LLM Models**
>
> **A4:** The main contribution of the paper is to present a novel method to extract geospatial knowledge from LLMs. It would be very interesting to determine whether or not the additional domain specific pre-training and finetuning of geospatial LLMs such as K-2 embeds additional knowledge that can be leveraged by GeoLLM. However, we believe that this investigation would be orthogonal from the main purpose of GeoLLM, which was simply to show that LLMs are new covariates and to present a method to use them as such. We leave this investigation to future work.

---

> ### Author Response · Authors · 2023-11-22
> **Reminder for Reviewer 92yi**
>
> Dear reviewer,
>
> As we are approaching the discussion deadline, we hope that your concerns are resolved and that you can reconsider our scores. Thank you very much for your time.

---

### Official Review · Reviewer_Gcfb · 2023-10-30

**Soundness:** 2 fair
**Presentation:** 3 good
**Contribution:** 3 good
**Rating:** 5
**Confidence:** 5

**Summary:**

This paper introduces a new method called GEOLLM, which can effectively extract spatial knowledge from large language models and fine tune it by using auxiliary map data from OpenStreetMap. Experiments on various tasks of several large-scale real-world sets show that the method is practical and effective. In the experiment, the author found that the pre-training language model has rich geospatial knowledge, and their method can unlock this knowledge. In addition, the author also discusses how to construct appropriate hints to extract geospatial knowledge, and how to find a balance between knowledge extraction and sample efficiency.
The contributions of the paper include demonstrating the quantity, quality, and scalable nature of geospatial knowledge contained in LLMs and presenting a simple and efficient method for extracting this knowledge.

**Strengths:**

S1: This paper delves into the internal structure of large-scale language models in extracting geographical knowledge. Utilizing auxiliary map data from OpenStreetMap to extract geospatial knowledge from large language models represents a new approach that has been sparingly explored in previous research. By fine-tuning basic models of different structures and scales and employing prompt templates containing addresses and nearby places, geographical knowledge within pre-trained models is effectively extracted.
S2: In terms of significance, this paper addresses a challenge in natural language processing by exploring how to extract geospatial knowledge from large-scale language models. The paper proposes a new approach to knowledge extraction based on fine-tuning and prompt strategies, establishing a new concept of "geospatial covariates."

**Weaknesses:**

W1: Lack of clear prompt templates and answer pairs design. The paper mentions the use of multiple data sources and various tasks; however, there are significant deficiencies in the theoretical approach. Specifically, the paper does not clearly specify whether the prompt formats are the same for each task and lacks detailed descriptions of how prompt templates and answer pairs were designed for different tasks. This lack of clarity makes it challenging for readers to understand the experimental design and poses obstacles for replication and further research.
W2: Lack of clear experimental design details. Furthermore, in terms of experimental design, there are shortcomings in this paper. It does not provide a clear overview of the experimental training process, including the selection of data, preprocessing steps, and hyperparameter tuning. This omission makes it difficult to evaluate the detailed experimental procedure, hindering the reproducibility of the experiments.
W3: Absence of comparison with ground-truth. Additionally, the paper does not thoroughly explore the comparison between experimental results and real-world geospatial data (Ground Truth). Accuracy of geospatial knowledge is crucial in practical applications; however, the paper lacks comparative analysis with actual geographical data.

**Questions:**

Besides what is mentioned in weaknesses, what does “the prompts’ mean in “This suggests that the model is less prone to overfitting on the prediction tasks compared to the prompts” on page 4?

---

> ### Author Response · Authors · 2023-11-20
> **Response to Reviewer Gcfb**
>
> Thank you for your insightful review and the recognition of the novel approach and significance of GeoLLM in extracting geographical knowledge from large language models using OpenStreetMap data. We appreciate your constructive feedback on the areas needing clarification and improvement. Below, we address your concerns and questions:
>
> **Q1: Lack of Clear Prompt Templates and Answer Pairs Design**
>
> **A1:** We added Appendix A.1 to provide details on the prompt template that we used across all tasks as well as an additional prompt example that we use in the asset wealth task. You can find details for the source of the auxiliary data used to construct the prompts in section 3.2. We hope this aids in the replicability of our results.
>
> **Q2: Lack of Detailed Experimental Design**
>
> **A2:** We added Appendix A.2 to describe the finetuning process and the exact hyperparameters used. Details on the preprocessing of the data for prompts are provided in the last paragraph of section 3.1. Details on the ground truth data for tasks, such as their sources or any task-specific preprocessing that we do, are provided in section 4.1.
>
> **Q3: Absence of Comparison with Ground-Truth Data**
>
> **A3:** Pearson’s $r^2$, which is the correlation squared, is shown for every model across all tasks in table 1. These are calculated using the ground truth and the predictions. We also show the mean absolute errors in an identically structured table in Appendix A.3. We clarify the fine-tuning process of the LLMs in Appendix A.2 where we describe how we use unsupervised learning on the prompts and labels (ground truth) concatenated. We also try to use the word “labels” more to be more clear about the fine-tuning process and the fact that we use ground truth. Please let us know if this is a sufficient clarification or if we are misunderstanding your point.
>
> **Q4: Clarification on the Statement Regarding Overfitting**
>
> **A4:** We clarify in section 3.3 and Appendix A.2 that we are using unsupervised learning with the prompts and labels concatenated. While the model may overfit (increase in loss on test set) with this unsupervised learning at the last few epochs, it does not seem to be overfitting on the actual “labels” or ground truth that were concatenated which is indicated by the continued improvement in performance on the test sets.
>
> We encourage you to see our overall response to see how we have strengthened our paper further or to answer any more questions you have.

---

> ### Author Response · Authors · 2023-11-22
> **Reminder for Reviewer Gcfb**
>
> Dear reviewer,
>
> As we are approaching the discussion deadline, we hope that your concerns are resolved and that you can reconsider our scores. Thank you very much for your time.

---

### Official Review · Reviewer_tFFk · 2023-10-30

**Soundness:** 3 good
**Presentation:** 3 good
**Contribution:** 3 good
**Rating:** 8
**Confidence:** 4

**Summary:**

This paper proposes GeoLLM, a prompt method that enriches geographic coordinates with auxiliary map data for geospatial prediction task (e.g., population density, economic livelihoods). This prompt approach outperforms some baseline methods in several tasks.

**Strengths:**

1.  This paper is well structured and good at the clarity of presentation.

2.  The significance of the problem is high, potentially impacting a wide range of geospatial applications and offering a new way to view large language model capabilities.

3. The paper provides a detailed discussion of how large language models can be linked to geospatial applications. This can potentially bring new research opportunities to the field of spatial data mining.

**Weaknesses:**

1.  The method presented seems to build incrementally on existing methodologies to generate auxiliary texts for prompt engineering, which may not represent a significance in technical novelty.

2.  The scope of experiments could be further expanded to support the conclusion. Please consider the Questions below for details.

**Questions:**

1.  While Peason's r^2 is used for comparisons with previous literature, can I see more comparisons with baselines on other evaluation metrics like MAE or MSE?

2.  Could the paper present the label distribution to evaluate performance variations between urban and rural areas, or densely versus sparsely populated regions?

3.  It would be beneficial to include a wider range of baselines, particularly "weaker" text embedding models like Doc2Vec, ELMo, and BERT, paired with basic predictors, for a holistic performance comparison.

4.  I tried to use GPT 4.0 for the example in Figure 1, and it looks like it can solve this question pretty well with only coordinates:
    """User: Coordinates:
    (40.76208, -73.98042)
    Population Density (On a Scale from 0.0 to 9.9):

    ChatGPT:
    Given the coordinates (40.76208, -73.98042), this location is in New York City, specifically in Manhattan. Manhattan has one of the highest population densities of any area in the United States.
    On a scale from 0.0 to 9.9, where 9.9 would represent the most densely populated, Manhattan would be close to the top of that scale. While I can't provide an exact number without more specific criteria for each point on the scale, it would be reasonable to rate Manhattan, particularly Midtown which is where these coordinates roughly point to, above 9.0 on a density scale capped at 9.9, considering its population density surpasses 10,000 people per square kilometer.

    User:
    Just return a number

    ChatGPT:
    9.5
    """
    The paper could enhance its soundness by examining the performance with more powerful LLMs. Performance insights using LLama 13b and 70b, as well as the latest GPT-4 model, would be valuable.




5.  Can I see the performance on LLMs without finetuning? It would be necessary to see the impact of fine-tuning in geospatial predictions.

6.  Few-shot learning capabilities are an intriguing aspect of LLMs. An analysis of the model’s few-shot performance, both pre and post fine-tuning, could significantly add to our understanding of the method's efficiency and flexibility.

7.  The current ablation study does not fully clarify the independent contribution of coordinates to the model's performance. Results from experiments that exclude coordinates from prompts would help demonstrate that the enhanced performance is not solely attributable to the auxiliary map data.

---

> ### Author Response · Authors · 2023-11-20
> **Response to Reviewer tFFk**
>
> Thank you for your positive evaluation of our paper, recognition of the clear presentation, and acknowledgment of the potential impact of our work in geospatial applications. We appreciate your constructive feedback and have addressed your concerns and questions as follows:
>
> **Q1: Incremental nature of methodology**
>
> **A1:** In addition to showing the importance of comprehensive prompts constructed from auxiliary map data, we revise the paper to show the importance of fine-tuning for overall performance. In particular we find that unsupervised learning (used for the pretraining of the LLMs) for fine-tuning works very well. The details of this finetuning for LLMs are included in section 3.3. Experiments comparing few-shot and finetuning are included in Appendix A.4.
>
> **Q2: Expansion of the scope of experiments**
>
> **A2:** With the revision, we add a sentence embeddings baseline, satellite imagery baseline, results for GPT-2 with GeoLLM, 3 new ablations, few-shot performance with GPT-3.5 and GPT-4, and a preliminary evaluation of the biases of LLMs. Please see “Zero-shot and few-shot experiments”, “Full set of ablations”, and “Preliminary investigation of bias” in the overall response.
>
> **Q3: Additional evaluation metrics (MAE, MSE)**
>
> **A3:** As a potentially more interpretable metric, we present the mean absolute error (MAE) for all models across all tasks and training sample sizes in Table 3 in Appendix A.3. One can observe that the same conclusions that are made with Pearson's $r^2$ can be made when comparing the MAE of the various models. In particular, the relative ranking in performance of the models within tasks is consistent with Pearson's $r^2$.
>
> **Q4: Performance variation in different geographical contexts**
>
> **A4:** While there does not appear to be any obvious signs of performance bias across countries or continents as seen in fig. 2, the existence of biases in performance is inevitable as the internet training corpora of LLMs are inherently biased towards developed and densely populated areas. We show preliminary signs of this bias in table 5 in Appendix A.5. We find that there is a moderate increase in mean absolute error between densely populated and sparsely populated areas and another moderate increase between the areas with above median asset wealth and below median asset wealth. As we discussed in section 5, this demonstrates that GeoLLM has the potential to be used as a tool to reveal LLM biases on a geographical scale. However, substantial additional research is required for a comprehensive analysis of the geospatial biases in LLMs and their training corpora.
>
> **Q5: Comparison with "weaker" text embedding models and a wider variety of baselines**
>
> **A5:** Please see “Stronger baselines” in the overall response.
>
> **Q6: Performance of more powerful LLMs**
>
> **A6:** Please see “Zero-shot and few-shot experiments” in the overall response.
>
> **Q7: Impact of fine-tuning on performance**
>
> **A7:** Please see “Zero-shot and few-shot experiments” in the overall response.
>
> **Q8: Few-shot learning capabilities**
>
> **A8:** Please see “Zero-shot and few-shot experiments” in the overall response.
>
> **Q9: Independent contribution of coordinates**
>
> **A9:** Please see “Full set of ablations” in the overall response.

---

> ### Author Response · Authors · 2023-11-22
> **Reminder for Reviewer tFFk**
>
> Dear reviewer,
>
> As we are approaching the discussion deadline, we hope that your concerns are resolved and that you can confirm that this is the case. Thank you very much for your time.

---

### Official Review · Reviewer_5MUc · 2023-10-31

**Soundness:** 3 good
**Presentation:** 3 good
**Contribution:** 2 fair
**Rating:** 6
**Confidence:** 3

**Summary:**

This study explores leveraging large language models (LLMs) for geospatial prediction tasks, addressing limitations of traditional covariates like satellite imagery. The authors introduce GeoLLM, an approach that effectively extracts geospatial knowledge from LLMs with auxiliary map data. The proposed method demonstrates a 70% improvement in performance compared to baselines, rivaling satellite-based benchmarks. GPT-3.5 outperforms other models, highlighting the scalability of the proposed approach. This research underscores LLMs' efficiency, global robustness, and potential to enhance geospatial analysis.

**Strengths:**

1. It proposed a novel method for efficiently extracting geospatial knowledge from large language models.
2. The paper outlined experiments to evaluate extracting geospatial knowledge from large language models, which included constructing a comprehensive benchmark, developing a robust set of baselines, and presenting results and an ablation study.
3. The paper revealed that GeoLLMs are sample-efficient, rich in geospatial information, and robust across the globe.

**Weaknesses:**

1. The paper does not provide a detailed analysis of the potential biases of LLMs and their training corpora.
2. It would be better to compare the GeoLLM’s performance with the results from satellite images.

**Questions:**

Have you tried the zero-shot or few-shot performance of various LLMs on the presented tasks?

---

> ### Author Response · Authors · 2023-11-20
> **Response to Reviewer 5MUc**
>
> Thank you for your thoughtful feedback, recognition of GeoLLM’s novelty and of our broad analysis across multiple relevant benchmark datasets with a comprehensive list of baselines and ablations. We appreciate your suggestions on investigating potential biases, few-shot or zero-shot performance, and on comparing with satellite images. You may find responses to your questions below:
>
> **Q1: Lack of a detailed analysis on biases of LLMs and training datasets**
>
> **A1:** While there does not appear to be any obvious signs of performance bias across countries or continents as seen in fig. 2, the existence of biases in performance is inevitable as the internet training corpora of LLMs are inherently biased towards developed and densely populated areas. We show preliminary signs of this bias in Table 5 in Appendix A.5. We find that there is a moderate increase in mean absolute error between densely populated and sparsely populated areas and another moderate increase between the areas with above median asset wealth and below median asset wealth. As we discussed in section 5, this demonstrates that GeoLLM has the potential to be used as a tool to reveal LLM biases on a geographical scale. However, substantial additional research is required for a comprehensive analysis of the geospatial biases in LLMs and their training corpora.
>
> **Q2: Comparison with methods that use satellite imagery**
>
> **A2:** Please see the Nightlight baseline in the “Stronger baselines” section of the overall response.
>
> **Q3: Including results from zero-shot or few-shot prompting**
>
> **A3:** Please see “Zero-shot and few-shot experiments” in the overall response.

---

> ### Author Response · Authors · 2023-11-22
> **Reminder for Reviewer 5MUc**
>
> Dear reviewer,
>
> As we are approaching the discussion deadline, we hope that your concerns are resolved and that you can reconsider our scores. Thank you very much for your time.

---

### Official Review · Reviewer_xypS · 2023-10-31

**Soundness:** 2 fair
**Presentation:** 2 fair
**Contribution:** 1 poor
**Rating:** 3
**Confidence:** 4

**Summary:**

This paper presents a method to extract prior geospatial knowledge from pretrained LLM.
The queried knowledge includes spatial demographics data, census data, and survey data.
The findings in the paper are interesting, however seem very preliminary. And the experiments are limited.

**Strengths:**

The paper is well presented, and the experiments cover several different geospatial datasets and tasks in relation to census and demographic data.

**Weaknesses:**

•	It seems that the proposed GeoLLM can only perform one specific task. After fine-tuning, is the fine-tuned LLMs (e.g., GPT-3.5) able to retain the ability to answer general questions that is not related to the specific task? Including some discussions about the generalization part could be useful.

•	The proposed model can only handle static/tabular geo information. It does not handle other types of spatial data, or spatiotemporal data and tasks.

•	Baselines are too simple. Considering some more recent deep-learning-based or Transformer-based baselines could be more convincing.

•	The experiments are limited.

**Questions:**

•	Is there any specific tokenization process introduced for GPS coordinates? Normally, the direct tokenizers with LLMs could split the GPS point into several different tokens, which might undermine the ability to understand GPS coordinates correctly.

•	Why using classification setting? The current loss functions that LLMs are using is a kind of binary, where only predictions that exactly match the masked word are considered correct and rewarded, while all other predictions are considered incorrect and penalized. So, there is no has no awareness and no sense of being close to or far away the correct answer. Normally, for the task described in this paper, this kind of sense should be valuable in getting better performance.

•	What about the deployment cost of the proposed GeoLLM? For example, compared to the baselines, what are the fine-tuning costs and inference costs of using GPT-3.5 or other LLMs. From my experience, fine-tuning GPT-3.5 can be very expensive.

---

> ### Author Response · Authors · 2023-11-20
> **Response to Reviewer xypS**
>
> Thank you for your thoughtful feedback, recognition of GeoLLM’s evaluation across a broad set of relevant geospatial datasets and tasks, and suggestions for clarifying our design choices. You may find responses to your concerns below:
>
> **Q1: GeoLLM performs a specific task/ does it answer general questions?**
>
> **A1:** GeoLLM is not intended to retain natural language or general-purpose response capabilities. It is a method to introduce a novel set of covariates that can enhance various downstream geospatial ML tasks that currently depend on traditional covariates such as satellite images. These tasks include poverty estimation, food security, biodiversity preservation, and environmental conservation. Thus, we can quickly adapt powerful LLMs to a variety of geospatial tasks using our novel prompting strategy. Moreover, we demonstrate that a high level of performance is achievable with relatively cheap finetuning using methods such as QLoRA.
>
> **Q2: Can GeoLLM handle different types of spatial/spatio-temporal data**
>
> **A2:** As LLMs have not yet been shown to be effective covariates for geospatial ML tasks, we focus on providing a method that enables this by efficiently extracting their spatial knowledge. In fact, there are two general settings for spatial prediction: location modeling and spatial contextual modeling [1]. The former aims at predicting the attributes/characteristics of a place given its current location. The latter focuses on predicting the attributes/characteristics of a place given its location as well as its spatial context. Our GeoLLM is able to handle both types of tasks. The spatial context modeling corresponds to our current GeoLLM setting since we give the location and description of the current location as well as its nearby places extracted from OpenStreetMap (See Figure 1 (b) bottom). The location modeling corresponds to the setting in which we delete the “Nearby place” in our prompt. We believe we are the first one that shows LLMs’ abilities on a wide range of geo-tasks. We acknowledge that there are other geo-tasks such as spatial relation prediction between two places, or questions requiring spatial footprints more than points (e.g., polylines and polygons). We treat them as future work.
>
> **Q3: Simplicity of baselines**
>
> **A3:** Please see “Stronger baselines” in the overall response.
>
> **Q4: Limited experiments**
>
> **A4:** Please see “Stronger baselines”,  “Zero-shot and few-shot experiments”, and “Full set of ablations” in the overall response.**
>
> **Q5: Tokenizer for geospatial coordinates**
>
> **A5:** We recognize the tokenization problem of the current LLM for geo-coordinates.  Current LLMs break up the GPS coordinates into multiple tokens and makes understanding them more difficult, especially for smaller LLMs like GPT-2. While larger models such as GPT-3.5 are better able to utilize the numeric value of the coordinates, as shown in the “only coordinates'' ablation in Table 2, the coordinates do not affect the final performance of GPT-3.5 significantly (bump from 0.72 to 0.73 Pearson’s $r^2$) as shown in the “no coordinates'' ablation in table 2.  In fact, GeoLLM is motivated by this observation and is developed to overcome this limitation. How to develop a tokenizer that makes LLM better understand geo-coordinate is a very interesting research direction but is orthogonal to our GeoLLM framework. We leave it as one of our future work.
>
> **Q6: Relevance of framing as a classification problem**
>
> **A6:** Making geospatial predictions via the generation of tokens is significantly more difficult than regression. As one would expect, GPT-2 struggles to perform well within this paradigm, even doing much worse than MLP-BERT that uses regression. For this very reason, we deliberately introduced decimals in the labels (0.0 to 9.9 as opposed to 0 to 99) to add consistency to the predictions as they always require exactly 3 tokens to generate (e.g. “5”, “.”, “3”). However, to our surprise, the autoregressive nature of LLMs not seem to hinder GPT-3.5 and Llama 2 significantly, with GPT-3.5 exceeding the performance of all models that use regression, including RoBERTa and the newly introduced MLP-BERT and Nightlight baselines. In addition, as GPT-3.5 only allows for fine-tuning through OpenAI’s API, we cannot make any modifications to the tokenization or architecture of the best performing model by far.
>
> **Q7: Fine-tuning and inference costs**
>
> **A7:** We present the costs of the finetuning of the LLMs we use in Appendix A.2. In short, fine-tuning GPT-3.5 with 10,000 samples costs 33 dollars. Thanks to its sample efficiency, one can finetune it with 1,000 samples for 5 dollars without much sacrifice to the overall performance. As a set of predictions only needs to be made once or a small number of times for a specific application, the deployment cost is not as significant or as much of a burden as with general-purpose LLMs.

---

> ### Author Response · Authors · 2023-11-20
> **Response to Reviewer xypS (Continued)**
>
> References:
>
> [1] Mai, Gengchen, Krzysztof Janowicz, Bo Yan, Rui Zhu, Ling Cai, and Ni Lao. "Multi-Scale Representation Learning for Spatial Feature Distributions using Grid Cells." In International Conference on Learning Representations. 2020.

---

> ### Author Response · Authors · 2023-11-22
> **Reminder for Reviewer xypS**
>
> Dear reviewer,
>
> As we are approaching the discussion deadline, we hope that your concerns are resolved and that you can reconsider our scores. Thank you very much for your time.

---

### Author Response · Authors · 2023-11-20
**Overall Author Response to Reviews**

We thank all the reviewers for their helpful feedback and suggestions. We appreciate that the reviewers recognize GeoLLM’s novelty in extracting geospatial information from LLMs (5MUc, Gcfb, 92yi), the comprehensive experimental evaluation of GeoLLM’s strong performance across many relevant global datasets (xypS, 5MUc, 92yi), and the significance of presenting an alternative to traditional geospatial covariates such as satellite images (tFFk, Gcfb, 92yi). All reviewers made helpful suggestions that strengthened our paper significantly.

The main contribution of this paper is to offer LLMs as a novel set of geospatial covariates that can enhance geospatial ML tasks in various domains such as poverty estimation, food security, biodiversity preservation, and environmental conservation.

We have addressed main concerns below:

1. **Stronger baselines** (xypS, 5MUc, tFFk, 92yi)

We have introduced 2 new baselines and an additional model that we think serves as a baseline for LLM performance with GeoLLM. Details for these new models are in Sections 4.2 and 3.3. The Pearon’s $r^2$ and mean absolute error (MAE) of all models across all tasks is shown in Table 1 and Table 3 respectively.

The first baseline, MLP-BERT, embeds the same prompt that is passed to GeoLLM through a (frozen) BERT model. All numerical and categorical features of the prompt as well as the embedding of the whole prompt from BERT are then fed to a 2-layer (trainable) MLP, which outputs the final prediction. While this baseline approaches the performance of RoBERTa, Llama 2 and GPT-3.5 both outperform this model on all tasks by a wide margin.

The second baseline, Nightlight, uses publicly available satellite images of luminosity at night (“nightlights”) from VIIRS which comes at a resolution of 500 meters. Nightlight images are commonly used as covariates as they are a proxy for economic development. We find that images of sizes 16 km by 16 km work well for our diverse set of tasks. We use gradient boosting trees (GBTs) as they have been shown to be effective for single-band nightlight imagery (Perez et al., 2017). We find that pretrained ResNet models, as used in (Yeh et al., 2020), require large numbers of samples to outperform GBTs. Somewhat predictably, Nightlight is only particularly strong on the population tasks. Notably, LLMs seem to be far more robust on a wide variety of tasks with GPT-3.5 and Llama 2 outperforming it on all tasks.

A model we add to our main results using GeoLLM is the 117 million parameter GPT-2. We use the same finetuning procedure using unsupervised learning as Llama 2. The inference is also done the same as the other LLMs. This model provides insight into the minimal performance attainable by an LLM and the actual difficulty of the task as an autoregressive model. Its underwhelming performance compared to Llama 2 and GPT-3.5 emphasizes the importance of the scale of the LLM and its dataset. It demonstrates the inherent disadvantage that LLMs have compared to the models and baselines that use regression.

2. **Zero-shot and few-shot experiments** (5MUc, tFFk, 92yi)

We show GPT-3.5 and GPT-4 few-shot performance in Table 4 in Appendix A.4. The main issue of using chat-based LLMs without finetuning is that they frequently refuse to give an answer, especially at remote or less developed locations. While zero-shot is unreliable, few-shot is far more consistent and controllable. We demonstrate their few-shot performance by providing them with a series of the 10 closest (by physical distance) training samples for each test example. To further prevent the models from refusing to answer, we use system messages “You are a detailed and knowledgeable geographer” and “You complete sequences of data with predictions/estimates” before providing the 10 examples. While few-shot does work, it seems to perform significantly worse than the same model fine tuned. In table 4, we see that a fine tuned GPT-3.5 outperforms GPT-3.5 and even GPT-4 by a wide margin.

3. **Full set of ablations**  (xypS, tFFk, 92yi)

We have added 3 new ablations to Table 2. This results in a complete set of the possible ablations to delineate which parts of the prompts are contributing to the overall performance. While we find that all components of the prompt contribute to the overall performance, the address and list of nearby places components individually contribute far more than the coordinates, which is especially the case with the larger LLMs like GPT-3.5.

We also Include a **preliminary analysis on the biases** of LLMs in Appendix A.5 (5MUc, tFFk), **mean absolute error (MAE) performance** across all models and tasks in Appendix A.3 (tFFk), details of the **prompt template** used across all tasks in Appendix A.1 (Gcfb), and **details on fine-tuning and hyperparameters** for the LLMs in Appendix A.2 (Gcfb).

Revision of the paper and new appendix: https://openreview.net/pdf?id=TqL2xBwXP3. Additions are marked in blue for easier visibility.

---

### Meta-Review · Area_Chair_U53z · 2023-12-09

**Metareview:**

A method is proposed to extract geospatial information contained in LLMs by fine-tuning on prompts constructed with auxiliary OSM data. The key strength of the paper is the demonstration of a simple but effective prompting strategy on several important geospatial tasks related to demographic, economic, or climate studies. A key weakness is the limited variety of prompt templates considered.

There is a variation in the reviewer opinions and detailed author feedback is provided to address them. Reviewer xypS questions the number of tasks handled and the extent of experimentation, where the rebuttal explains the variety of tasks and adds more experiments. Reviewer Gcfb questions comparison to ground truth, which is addressed by the rebuttal in pointing out Pearson’s r^2 estimated for various models, while MAE and MSE are added in response to Reviewer tFFk who is satisfied with the additional metrics and experiments. More experiments requested by Reviewer 5MUc are added to study LLM biases and add comparisons to satellite images. Reviewer 92yi requires comparison to a stronger MLP-BERT baseline, which is provided. While the comparison with fine-tuning other geospatial LLM models is not implemented, Reviewer 92yi is satisfied with the set of contributions. Overall, the AC has carefully considered the paper, reviews and author feedbacks, to determine that all major concerns have been sufficiently addressed, thus, the paper is recommended for acceptance to ICLR.

**Justification For Why Not Higher Score:**

While experiments are comprehensive, some alternatives such as other prompt templates and other geospatial LLMs are not considered.

**Justification For Why Not Lower Score:**

A simple but effective idea that works well across a range of interesting geospatial tasks.

---

### Decision · Program_Chairs · 2024-01-16

Accept (poster)